# Enhanced ER-associated degradation of HMG CoA reductase causes embryonic lethality associated with *Ubiad1* deficiency

**Youngah Jo[1], Steven S Kim[1], Kristina Garland[1], Iris Fuentes[1], Lisa M DiCarlo[1], Jessie L Ellis[2], Xueyan Fu[2], Sarah L Booth[2], Bret M Evers[3], Russell A DeBose-Boyd[1]\***

[1]Department of Molecular Genetics, University of Texas Southwestern Medical, Dallas, United States; [2]Center at Dallas and Jean Mayer USDA Human Nutrition Research Center on Aging, Tufts University, Somerville, United States; [3]Department of Pathology, University of Texas Southwestern Medical, Dallas, United States

**Abstract** UbiA prenyltransferase domain-containing protein-1 (UBIAD1) synthesizes the vitamin K subtype menaquinone-4 (MK-4). Previous studies in cultured cells (Schumacher et al., 2015) revealed that UBIAD1 also inhibits endoplasmic reticulum (ER)-associated degradation (ERAD) of ubiquitinated HMG CoA reductase (HMGCR), the rate-limiting enzyme of the mevalonate pathway that produces cholesterol and essential nonsterol isoprenoids. Gene knockout studies were previously attempted to explore the function of UBIAD1 in mice; however, homozygous germ-line elimination of the *Ubiad1* gene caused embryonic lethality. We now report that homozygous deletion of *Ubiad1* is produced in knockin mice expressing ubiquitination/ERAD-resistant HMGCR. Thus, embryonic lethality of *Ubiad1* deficiency results from depletion of mevalonate-derived products owing to enhanced ERAD of HMGCR rather than from reduced synthesis of MK-4. These findings provide genetic evidence for the significance of UBIAD1 in regulation of cholesterol synthesis and offer the opportunity in future studies for the discovery of new physiological roles of MK-4.

**\*For correspondence:**
Russell.Debose-Boyd@
utsouthwestern.edu

**Competing interests:** The authors declare that no competing interests exist.

## Introduction

Vitamin K refers to a group of lipophilic molecules that serve as a cofactor for γ-carboxyglutamyl carboxylase, which converts specific glutamate residues in a limited set of proteins to γ-carboxyglutamate (*Shearer and Newman, 2014*; *Shearer and Okano, 2018*). This post-translational modification is obligatory for biological functions of resultant vitamin K-dependent proteins (VKDPs), some of which play key roles in blood coagulation. Other VKDPs are implicated in processes ranging from bone and cardiovascular mineralization to energy metabolism and inflammation (*Booth, 2009*; *Shearer and Okano, 2018*). In addition, vitamin K may exert direct effects on gene expression, signal transduction, and cellular regulation.

All vitamin K forms include a common 2-methyl-1,4-naphthoquinone ring structure known as menadione (MD) (*Figure 1A*) and are distinguished from one another by length and saturation of the side chain attached at the 3-carbon position on the ring (*Shearer and Newman, 2014*). MD is a provitamin form of vitamin K as the side chain is required for vitamin K activity (*Buitenhuis et al., 1990*). Phylloquinone (PK, also known as vitamin K$_1$) (*Figure 1A*) contains a phytyl side chain, whereas menaquinones (MKs, collectively referred to as vitamin K$_2$) contain a side chain with 5-carbon isoprenyl units and are named according to the number of these units (e.g., MK-n) (*Figure 1A*). PK is

produced by plants, whereas longer chain MKs (MK-7, MK-9, and MK-11) are predominantly of bacterial origin. Invertebrate and vertebrate animals produce MK-4 from dietary PK through a reaction involving side chain removal and re-addition with MD as an intermediate (*Al Rajabi et al., 2012*; *Okano et al., 2008*).

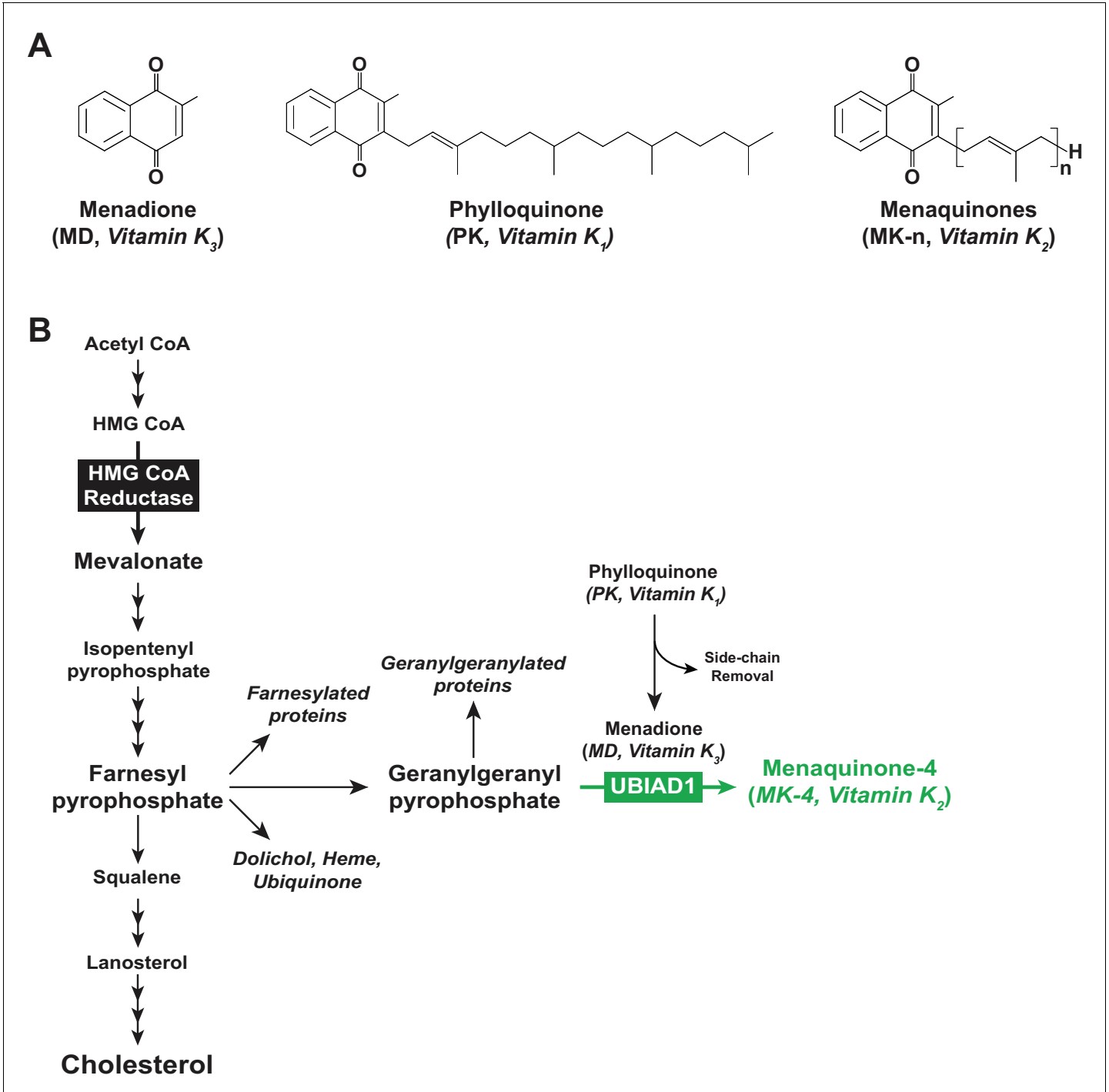

**Figure 1.** Vitamin K and the mevalonate pathway. (**A**) Structures of the main forms of vitamin K. (**B**) The mevalonate pathway in animal cells.

UbiA prenyltransferase domain-containing protein-1 (UBIAD1), a member of the UbiA superfamily of prenyltransferases (*Li, 2016*), transfers the 20-carbon geranylgeranyl group from geranylgeranyl pyrophosphate (GGpp) to PK-derived MD, thereby producing MK-4 (*Hirota et al., 2013*; *Nakagawa et al., 2010*; *Figure 1B*). The function of UBIAD1 appears to extend beyond its role in MK-4 synthesis, as indicated by association of mutations in human *UBIAD1* with Schnyder corneal dystrophy (SCD) (*Orr et al., 2007*; *Weiss et al., 2007*). This rare, autosomal-dominant disease is characterized by progressive corneal opacification that results from accumulation of cholesterol. In 2015, we showed that SCD-associated UBIAD1 inhibits the sterol-accelerated, endoplasmic reticulum (ER)-associated degradation (ERAD) of 3-hydroxy-3-methylglutaryl coenzyme A reductase (HMGCR) (*Schumacher et al., 2015*), one of several feedback mechanisms that converge on the enzyme to assure cholesterol homeostasis (*Brown and Goldstein, 1980*).

Polytopic, ER-localized HMGCR produces mevalonate, an important intermediate in synthesis of cholesterol and the nonsterol isoprenoids farnesyl pyrophosphate (Fpp) and GGpp that are transferred to many cellular proteins and utilized in synthesis of other nonsterol isoprenoids including MK-4, heme, ubiquinone-10, and dolichol (*Goldstein and Brown, 1990*; *Wang and Casey, 2016*) (see *Figure 1B*). Sterols accelerate ERAD of HMGCR by stimulating its binding to ER membrane proteins called Insigs (*Sever et al., 2003a*; *Sever et al., 2003b*). Insig-associated ubiquitin ligases facilitate ubiquitination of HMGCR (*Jiang et al., 2018*; *Jo et al., 2011*; *Song et al., 2005*), marking it for extraction across ER membranes and subsequent cytosolic release for ERAD by 26S proteasomes (*Morris et al., 2014*). ERAD of HMGCR is enhanced by GGpp, which enhances membrane extraction of the ubiquitinated enzyme (*Elsabrouty et al., 2013*).

Sterols also cause HMGCR to bind UBIAD1 (*Schumacher et al., 2015*). This binding inhibits the ERAD of HMGCR at a post-ubiquitination step in the reaction, thereby permitting continued synthesis of mevalonate for incorporation into nonsterol isoprenoids even when intracellular sterols are abundant (*Schumacher et al., 2018*). GGpp triggers release of UBIAD1 from HMGCR, which allows for maximal ERAD of HMGCR and translocation of UBIAD1 from the ER to the *medial-trans* Golgi. SCD-associated mutations cluster around the membrane-embedded active site of UBIAD1 (*Cheng and Li, 2014*; *Huang et al., 2014*), indicating they may disrupt sensing of GGpp. Indeed, SCD-associated UBIAD1 is refractory to GGpp-induced release from HMGCR and becomes sequestered in the ER (*Schumacher et al., 2016*). The resultant inhibition of HMGCR ERAD leads to enhanced synthesis and intracellular accumulation of cholesterol (*Schumacher et al., 2018*).

To explore the in vivo function of UBIAD1, efforts were attempted to generate mice lacking *Ubiad1* (*Nakagawa et al., 2014*). However, mouse embryos homozygous for *Ubiad1* deficiency failed to survive past embryonic day 7.5. We recently observed that the ERAD of HMGCR was enhanced in transformed human fibroblasts lacking UBIAD1 (*Schumacher et al., 2018*). This observation led us to speculate that embryonic lethality of *Ubiad1* deficiency in mice results from mevalonate depletion due to accelerated ERAD of HMGCR rather than from reduced synthesis of MK-4. We evaluate this notion here by determining whether ubiquitination/ERAD-resistant HMGCR rescues embryonic lethality of *Ubiad1*-deficiency.

## Results

We used CRISPR/Cas9 methods to introduce heterozygous *Ubiad1* deficiency in wild type (WT) and previously described *Hmgcr*$^{Ki/Ki}$ mice (*Hwang et al., 2016*), which harbor knockin mutations that prevent ubiquitination and subsequent ERAD of HMGCR (*Sever et al., 2003a*). These mice are designated *Ubiad1*$^{+/-}$ and Ubiad1$^{+/-}$: :HmgcrKi$^{Ki/Ki}$. Two independent lines of mice were obtained in which the *Ubiad1* gene was disrupted by a 172- (Disrupted Allele A) or 29 bp deletion (Disrupted Allele B) in exon 1 (*Figure 2*). If transcribed and translated, these alleles would produce protein fragments comprising amino acids 1–38 (Disrupted Allele B) or 1–39 (Disrupted Allele A) of UBIAD1 fused to a novel polypeptide of 55 or 56 amino acids (*Figure 2*). *Table 1* shows results of breeding experiments in which mice heterozygous for *Ubiad1* deletion were mated and genotypes of offspring determined by PCR analysis. Intercrosses of *Ubiad1*$^{+/-}$ mice produced WT and *Ubiad1*$^{+/-}$ offspring at a ratio of approximately 1:2, which is consistent with Mendelian segregation. However, *Ubiad1*$^{-/-}$ offspring were not produced, regardless of disrupted *Ubiad1* allele. In striking contrast, all three expected genotypes (*Ubiad1*$^{+/+}$: :*Hmgcr*$^{Ki/Ki}$, *Ubiad1*$^{+/-}$: :*Hmgcr*$^{Ki/Ki}$, *and Ubiad1*$^{-/-}$: :*Hmgcr*$^{Ki/Ki}$) were produced when *Ubiad1*$^{+/-}$: :*Hmgcr*$^{Ki/Ki}$ mice were intercrossed (*Table 1*). The observed

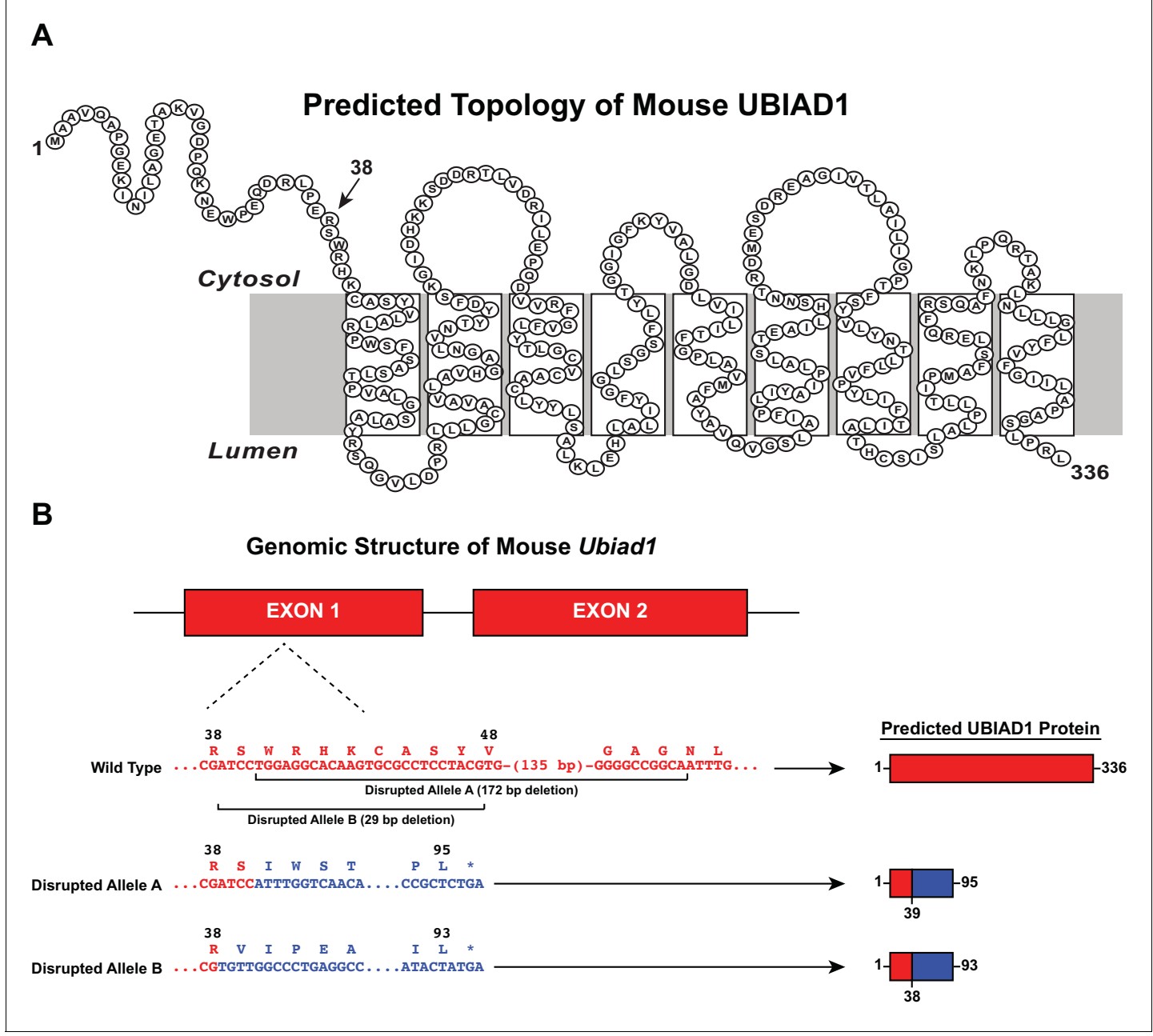

**Figure 2.** CRISPR/Cas9-mediated disruption of the mouse *Ubiad1* gene. (**A**) Amino acid sequence and predicted topology of mouse UBIAD1 protein. (**B**) Genomic structure of mouse *Ubiad1* and predicted proteins encoded by CRISPR/Cas9-disrupted *Ubiad1* alleles (Disrupted Alleles A and B).

proportion of +/+ : +/- : -/- *Ubiad1* alleles was 137:280:114 (Disrupted Allele A) and 23:36:20 (Disrupted Allele B), when the expected proportion is 1:2:1. These results provide genetic evidence that ubiquitination/ERAD-resistant HMGCR rescues embryonic lethality of *Ubiad1* deficiency.

In *Figure 3A*, we measured UBIAD1 and HMGCR protein levels in livers of 8-week-old WT and *Ubiad1*[+/-] mice. Male and female *Ubiad1*[+/-] mice exhibited reduced levels of hepatic UBIAD1 protein as expected (*Figure 3A*, compare lanes 1 and 3 with lanes 2 and 4). HMGCR protein was slightly reduced in livers of the mice (lanes 1–4), which likely resulted from enhanced ERAD. Similar results were obtained with livers of mice harboring Disrupted Allele B (data not shown). Sterol regulatory element-binding protein (SREBP)−1 and −2 are transcription factors synthesized as inactive, ER-bound precursors (*Brown and Goldstein, 1997*). Upon lipid deprivation, transcriptionally active

**Table 1.** Segregation of Disrupted *Ubiad1* Alleles in Mice.

| Genotype of breeding pairs | *Ubiad1* genotype of offspring | | |
| --- | --- | --- | --- |
| | +/+ | +/- | -/- |
| Disrupted Allele A<br>*Ubiad1*$^{+/-}$ X *Ubiad1*$^{+/-}$ | 83 | 201 | 0 |
| *Ubiad1*$^{+/-}$: :HmgcrKi$^{Ki/Ki}$<br>X<br>*Ubiad1*$^{+/-}$: :HmgcrKi$^{Ki/Ki}$ | 137 | 280 | 114 |
| Disrupted allele B<br>*Ubiad1*$^{+/-}$ X *Ubiad1*$^{+/-}$ | 77 | 183 | 0 |
| *Ubiad1*$^{+/-}$: :HmgcrKi$^{Ki/Ki}$<br>X<br>*Ubiad1*$^{+/-}$: :HmgcrKi$^{Ki/Ki}$ | 23 | 36 | 20 |

Genotype was determined by PCR analysis of genomic DNA prepared from tails of mice.

fragments of SREBPs are proteolytically released from membranes and migrate to the nucleus where they activate transcription of genes encoding cholesterol and fatty acid synthetic enzymes (*Horton et al., 2003*). The level of membrane-bound precursor and nuclear forms of SREBPs remained constant in *Ubiad1*$^{+/-}$ mice (lanes 5–8 and 9–12). Despite reduced levels of hepatic UBIAD1 and HMGCR protein, *Ubiad1*$^{+/-}$ mice were indistinguishable from WT littermates and had similar body weights.

The absence of UBIAD1 protein from hepatic membranes of *Ubiad1*$^{-/-}$: :*Hmgcr*$^{Ki/Ki}$ was confirmed in *Figure 3B* (compare lanes 1–2 and 4–5 with lanes 3 and 6, respectively). The amount of HMGCR protein and nuclear SREBPs was varied in livers of the animals (lanes 1–18); however, the nature of this variation was not clear. Although ubiquitination/ERAD-resistant HMGCR rescued embryonic lethality of *Ubiad1* deficiency, male and female *Ubiad1*$^{-/-}$: :*Hmgcr*$^{Ki/Ki}$ mice were smaller (30% and 20%, respectively) than their *Ubiad1*$^{+/+}$: :*Hmgcr*$^{Ki/Ki}$ and *Ubiad1*$^{+/-}$: :*Hmgcr*$^{Ki/Ki}$ littermates at 8 weeks of age. Similar results were observed with 8-week-old *Ubiad1*$^{-/-}$: :*Hmgcr*$^{Ki/Ki}$ mice harboring Disrupted Allele B (*Figure 3—figure supplement 1A*). Hepatic levels of cholesterol and triglycerides (*Figure 3—figure supplement 1B*) as well as most mRNAs encoding SREBPs, SREBP pathway components, and cholesterol/fatty acid synthetic enzymes were not globally changed in the absence of *Ubiad1* (*Figure 3—figure supplement 1C*). Notably, the mRNA encoding Insig-2a (the major Insig-2 isoform in the liver) was reduced, whereas the minor Insig-2b transcript was slightly increased in *Ubiad1*-deficient mice. The variation in Insig-2 mRNA, HMGCR, and nuclear SREBPs could be related to variations in food intake or failure of *Ubiad1*-deficient mice to thrive (see below). For all experiments described hereafter, male and female *Ubiad1*$^{+/-}$: :*Hmgcr*$^{Ki/Ki}$ mice with Disrupted Allele A were crossed to obtain *Ubiad1*$^{+/+}$: :*Hmgcr*$^{Ki/Ki}$, *Ubiad1*$^{+/-}$: :*Hmgcr*$^{Ki/Ki}$, and *Ubiad1*$^{-/-}$: :*Hmgcr*$^{Ki/Ki}$ littermates for analysis.

*Figure 3C* compares post-weaning weight gain of *Ubiad1*$^{+/+}$: :*Hmgcr*$^{Ki/Ki}$, *Ubiad1*$^{+/-}$: :*Hmgcr*$^{Ki/Ki}$, and *Ubiad1*$^{-/-}$: :*Hmgcr*$^{Ki/Ki}$ mice consuming chow diet ad libitum. The results show that *Ubiad1*$^{+/+}$: :*Hmgcr*$^{Ki/Ki}$ and *Ubiad1*$^{+/-}$: :*Hmgcr*$^{Ki/Ki}$ mice gained weight at similar rates up to ~8 weeks post-weaning. *Ubiad1*$^{-/-}$: :*Hmgcr*$^{Ki/Ki}$ mice gained weight up to 2 weeks post-weaning (albeit at a reduced rate compared to littermates), after which weight gain plateaued. After 7.7 weeks, male and female *Ubiad1*$^{-/-}$: :*Hmgcr*$^{Ki/Ki}$ mice were 30–40% smaller than *Ubiad1*$^{+/+}$: :*Hmgcr*$^{Ki/Ki}$ and *Ubiad1*$^{+/-}$: :*Hmgcr*$^{Ki/Ki}$ littermates.

The amount of UBIAD1 and HMGCR protein was next measured in various tissues of male *Ubiad1*$^{+/+}$: :*Hmgcr*$^{Ki/Ki}$ and *Ubiad1*$^{-/-}$: :*Hmgcr*$^{Ki/Ki}$ mice. As expected, UBIAD1 was not detected in membranes isolated from the liver, pancreas, brain, kidney, and spleen of *Ubiad1*-deficient mice (*Figure 4A*). *Figure 4B* shows that MK-4, the product of UBIAD1 enzymatic activity, accumulated to the highest level in the pancreas of *Ubiad1*$^{+/+}$: :*Hmgcr*$^{Ki/Ki}$ mice. Lower levels of MK-4 were found in the brain, kidney, spleen, and liver of the animals. In contrast, MK-4 failed to accumulate to detectable levels in tissues of *Ubiad1*-deficient mice.

Having established the absence of UBIAD1 protein and its enzymatic product (MK-4) in *Ubiad1*-deficient mice, we next compared blood chemistries between *Ubiad1*$^{+/+}$: :*Hmgcr*$^{Ki/Ki}$ and *Ubiad1*$^{-/-}$: :*Hmgcr*$^{Ki/Ki}$ animals. Serum cholesterol levels were not significantly different between the two groups

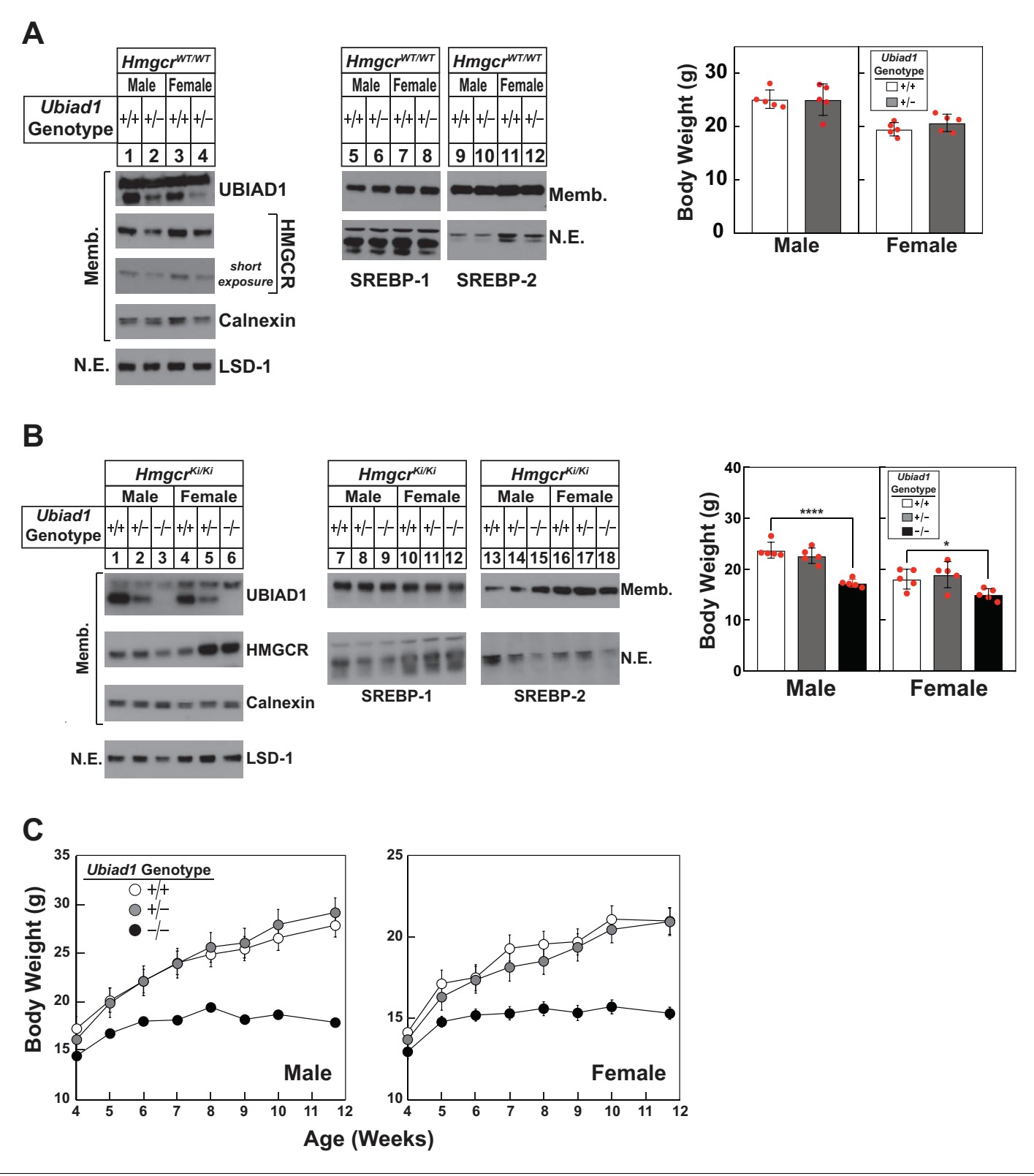

**Figure 3.** Hepatic immunoblot analysis and body weights of *Ubiad1*-deficient mice on *Hmgcr*^*WT/WT*^ and *Hmgcr*^*Ki/Ki*^ backgrounds. (A and B) Male and female WT and *Ubiad1*^*+/-*^ (A) or *Ubiad1*^*+/+*^: :*Hmgcr*^*Ki/Ki*^, *Ubiad1*^*+/-*^: :*Hmgcr*^*Ki/Ki*^, and *Ubiad1*^*-/-*^: :*Hmgcr*^*Ki/Ki*^ (B) littermates (8 weeks of age, five mice/ group) were fed an ad libitum chow diet prior to weighing and sacrifice. Livers were harvested and subjected to subcellular fractionation as described in 'Materials and methods.' Aliquots of resulting membrane (Memb.) and nuclear extract (N.E.) fractions (80–160 µg protein/lane) for each group were

*Figure 3 continued on next page*

*Figure 3 continued*

pooled and subjected to SDS-PAGE, followed by immunoblot analysis using antibodies against endogenous HMGCR, UBIAD1, SREBP-1, SREBP-2, calnexin, and LSD-1. Although shown in separate panels, LSD-1 is a loading control for nuclear SREBP immunoblots. (C) Male and female $Ubiad1^{+/+}$: : $Hmgcr^{Ki/Ki}$, $Ubiad1^{+/-}$: : $Hmgcr^{Ki/Ki}$, and $Ubiad1^{-/-}$: : $Hmgcr^{Ki/Ki}$ littermates (eight mice/group) were weaned at 4 weeks of age, fed chow diet ad libitum, and weighed for seven consecutive weeks, after which they were sacrificed. *Error bars*, S.E. *, p<0.05 and ****, p<0.0001.

The online version of this article includes the following source data and figure supplement(s) for figure 3:

**Source data 1.** Body weights of $Ubiad1^{-/-}$: : $Hmgcr^{Ki/Ki}$ mice.

**Figure supplement 1.** Characterization of *Ubiad1*-deficient mice.

**Figure supplement 1—source data 1.** Body weights and hepatic lipid levels of $Ubiad1^{-/-}$: : $Hmgcr^{Ki/Ki}$ mice.

---

of male mice (*Figure 4C*). Modest, but significant increases in levels of triglycerides (55%) and non-esterified fatty acids (46%) were observed in the serum of $Ubiad1^{-/-}$: :$Hmgcr^{Ki/Ki}$ mice compared to $Ubiad1^{+/+}$: :$Hmgcr^{Ki/Ki}$ littermates. The *Ubiad1*-deficient mice exhibited larger increases in serum levels of two classic markers of liver injury, alanine aminotransferase (ALT) (174%) and aspartate amino-transferase (AST) (583%); serum alkaline phosphatase (ALP) was reduced approximately 35%. Similar results were obtained with serum from female $Ubiad1^{-/-}$: :$Hmgcr^{Ki/Ki}$ mice (*Figure 4—figure supplement 1A*).

In *Figure 4D*, we conducted a second blood chemistry analysis on male $Ubiad1^{+/+}$: :$Hmgcr^{Ki/Ki}$ and $Ubiad1^{-/-}$: :$Hmgcr^{Ki/Ki}$ mice. Similar to results of *Figure 4C*, serum levels of AST and ALT were elevated 5-fold and 1.8-fold, respectively, in the absence of *Ubiad1* (*Figure 4D*). Serum lactate dehydrogenase (LDH) was elevated approximately 2-fold in *Ubiad1*-deficient mice; however, a more prominent elevation (7.5-fold) of serum creatine kinase (CK) was observed. Finally, a slight reduction in the amount of serum lipase and amylase was present in *Ubiad1*-deficient mice; serum albumin remained unchanged. Serum from female $Ubiad1^{-/-}$: :$Hmgcr^{Ki/Ki}$ mice exhibited similar characteristics (*Figure 4—figure supplement 1B*).

To further characterize $Ubiad1^{-/-}$: :$Hmgcr^{Ki/Ki}$ mice, we conducted a complete histological analysis of all tissues from the animals. Surprisingly, abnormalities were observed in only two tissues of *Ubiad1*-deficient mice – skeletal muscle and bone. Hematoxylin and eosin (H and E)-staining of gastrocnemius (*Figure 5A*, panels 1–4) and quadriceps muscles (panels 5–8) from male *Ubiad1*-deficient mice revealed occasional degenerating myofibers with macrophage infiltration as well as frequent myofibers with centrally-localized nuclei. Similar results were observed in gastrocnemius and quadriceps muscles isolated from female mice (*Figure 5—figure supplement 1*). Overall, these histological findings are indicative of ongoing muscle injury and correlate to the elevated serum CK levels observed in *Figure 4D*. We used H and E together with Safranin O staining to examine growth plates in femurs from both male (*Figure 5B*) and female (*Figure 5—figure supplement 2*) $Ubiad1^{+/+}$: :$Hmgcr^{Ki/Ki}$ and $Ubiad1^{-/-}$: :$Hmgcr^{Ki/Ki}$ mice. The results reveal that *Ubiad1* deficiency led to the disorganization of cells within proliferative and hypertrophic zones of the growth plate, persistence of cartilage within trabeculae, and a mild decrease in the number of boney trabeculae.

## Discussion

The genetic ablation of *UBIAD1* in transformed human fibroblasts led to enhanced ERAD of HMGCR and reduced cholesterol synthesis and intracellular accumulation of cholesterol (*Schumacher et al., 2018*). These observations prompted us to consider the possibility that embryonic lethality associated with *Ubiad1* deficiency in mice resulted from mevalonate depletion. Indeed, homozygous *Hmgcr* deficiency caused early embryonic lethality in mice (*Ohashi et al., 2003*), establishing that mevalonate-derived metabolites are crucial for embryonic development. We previously generated $Hmgcr^{Ki/Ki}$ mice, which harbor knockin mutations in the *Hmgcr* gene that prevent sterol-induced ubiquitination and subsequent ERAD of HMGCR (*Hwang et al., 2016*). As a result of resistance to ERAD, HMGCR protein accumulated in tissues of $Hmgcr^{Ki/Ki}$ mice that stimulated the overproduction of cholesterol and likely other sterol and nonsterol isoprenoids. Hence, we reasoned that overproduction of sterol and nonsterol isoprenoids in $Hmgcr^{Ki/Ki}$ mice would rescue embryonic lethality associated with *Ubiad1* deficiency. Our current studies show that homozygous *Ubiad1* deletion was produced at expected Mendelian ratios in $Hmgcr^{Ki/Ki}$, but not in WT mice (*Table 1*). This important observation not only highlights the crucial role for UBIAD1 in regulation of HMGCR ERAD, but also

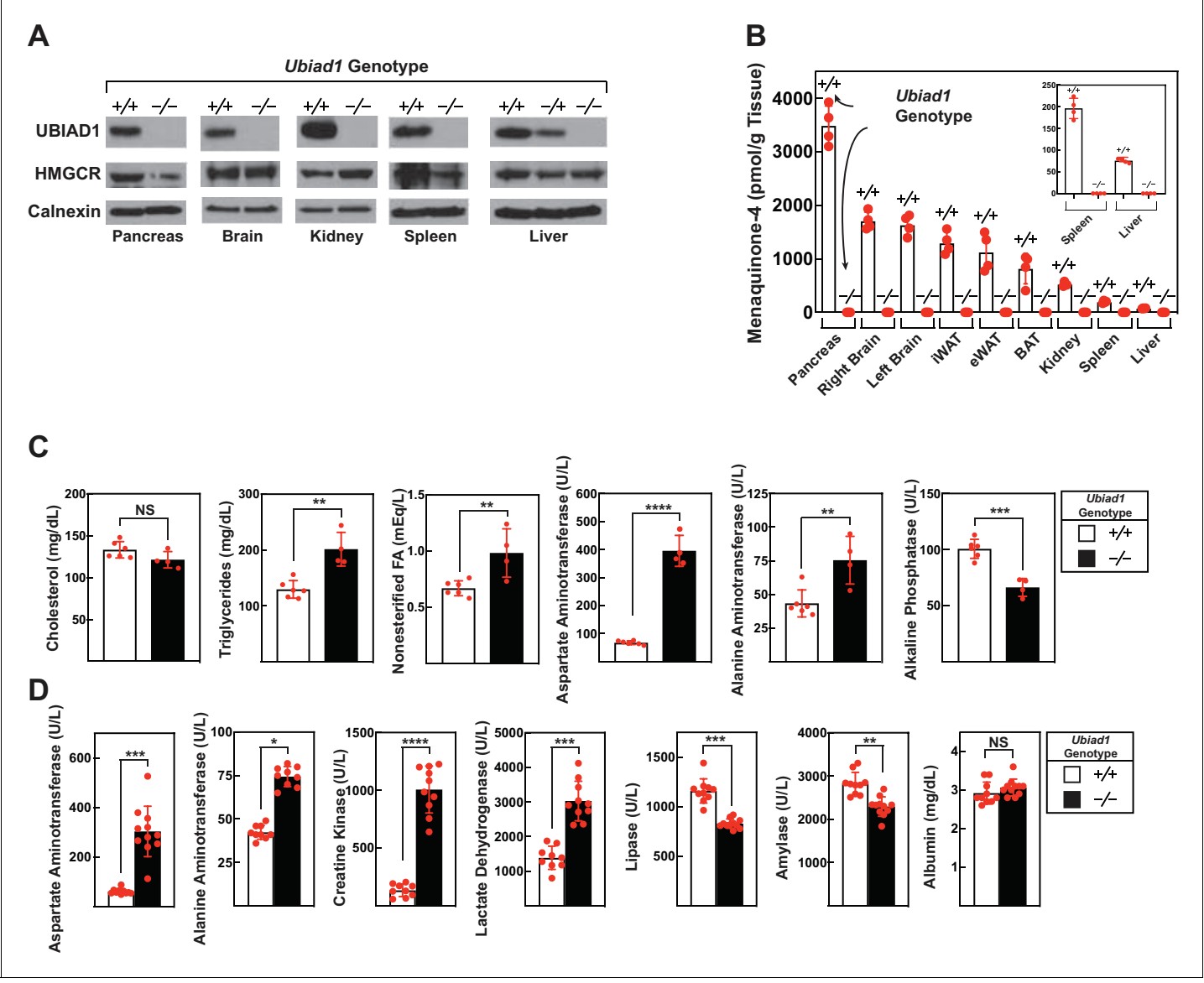

**Figure 4.** Levels of HMGCR, UBIAD1, and MK-4 in various tissues and blood chemistry analysis of *Ubiad1*-deficient mice. Male *Ubiad1*[+/+]*: :Hmgcr*[Ki/Ki] and *Ubiad1*[-/-]*: :Hmgcr*[Ki/Ki] littermates (12 weeks of age, 4–11 mice/group) were fed an ad libitum chow diet prior to sacrifice. (**A and B**) Indicated tissues were harvested for subcellular fractionation, after which aliquots of membrane fractions were subjected to immunoblot analysis using antibodies against HMGCR, UBIAD1, and calnexin (**A**). Some of the tissues were subjected to homogenization (**B**) for subsequent determination of MK-4 levels by reverse-phase high performance liquid chromatography or liquid chromatography-mass spectrometry as described in 'Materials and methods.' (**C and D**) Blood drawn from mice following sacrifice was subjected to chemical analysis by the Metabolic Phenotyping Core Facility in the Touchstone Diabetes Center (UT Southwestern Medical Center). *Bars*, mean ± S.E. of data from 4 to 11 mice. \*, p<0.05; \*\*, p<0.01; \*\*\*, p<0.001; \*\*\*\*, p<0.0001.
The online version of this article includes the following source data and figure supplement(s) for figure 4:

**Source data 1.** Blood chemistry analysis of male *Ubiad1*[-/-]*: : Hmgcr*[Ki/Ki] mice.
**Figure supplement 1.** Blood chemistry analysis of female *Ubiad1*[+/+]*: :Hmgcr*[Ki/Ki] and *Ubiad1*[-/-]*: :Hmgcr*[Ki/Ki] mice.
**Figure supplement 1—source data 1.** Blood chemistry analysis of female *Ubiad1*[-/-]*: : Hmgcr*[Ki/Ki] mice.

confirms that abrogating the reaction allows production of a mevalonate-derived metabolite(s) that rescues embryonic lethality associated with *Ubiad1* deficiency. Notably, administration of MK-4, ubiquinone-10 (*Nakagawa et al., 2014*), or cholesterol (data not shown) to *Ubiad1*[+/-] mice prior to intercrossing and throughout pregnancy failed to rescue embryonic lethality of *Ubiad1* deficiency. While the identity of the mevalonate-derived metabolite(s) that rescues embryonic development

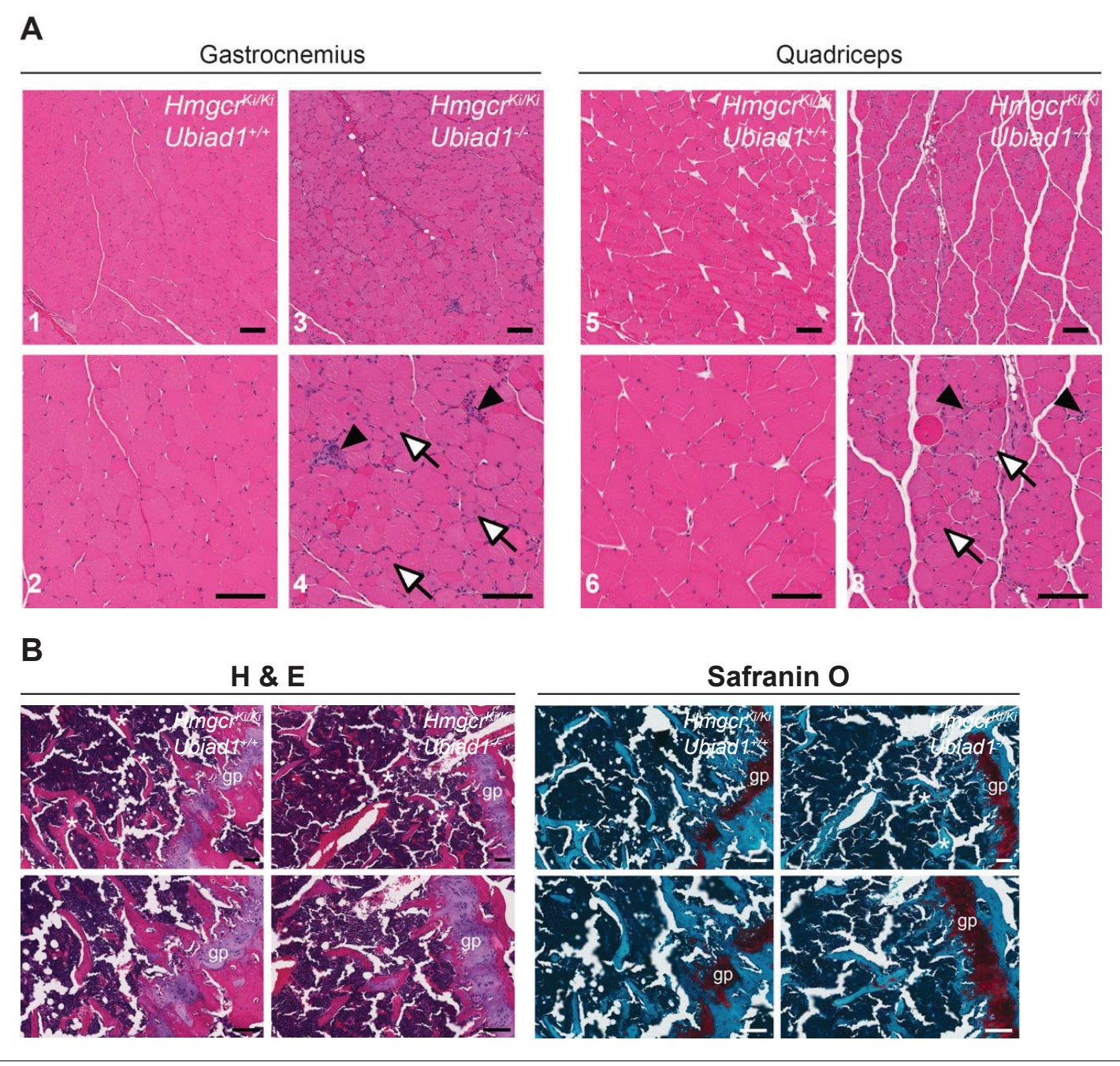

**Figure 5.** Histological analysis of skeletal muscle and femoral growth plates from *Ubiad1*[+/+]: *Hmgcr*[Ki/Ki] and *Ubiad1*[-/-]: *Hmgcr*[Ki/Ki] mice. Histological analysis of gastrocnemius and quadriceps muscles (**A**) and femoral growth plates (**B**) from male *Ubiad1*[+/+]: *Hmgcr*[Ki/Ki] and *Ubiad1*[-/-]: *Hmgcr*[Ki/Ki] littermates using H and E and Safranin O staining. Myofibers harboring centrally-localized nuclei are indicated by white arrows, and degenerating myofibers with macrophage infiltration are indicted by black arrowheads in (**A**). Asterisks in (**B**) indicate boney trabeculae, and red hue in Safranin O-stained sections highlight cartilage. gp, growth plate. Scale bars, 100 μm.

The online version of this article includes the following figure supplement(s) for figure 5:

**Figure supplement 1.** Histological analysis of gastrocnemius and quadriceps muscles from female *Ubiad1*[+/+]: *Hmgcr*[Ki/Ki] and *Ubiad1*[-/-]: *Hmgcr*[Ki/Ki] littermates using H and E staining.

**Figure supplement 2.** Histological analysis of femoral growth plates from female *Ubiad1*[+/+]: *Hmgcr*[Ki/Ki] and *Ubiad1*[-/-]: *Hmgcr*[Ki/Ki] littermates using H and E and Safranin O staining.

remains unknown, genes encoding HMGCR, enzymes required for GGpp and Fpp synthesis, and prenylation of small GTPases are essential for migration of primordial germ cells during embryonic development (*Kunwar et al., 2006*). Thus, depletion of GGpp and/or Fpp owing to enhanced ERAD of HMGCR and reduced prenylation of small GTPases may contribute to embryonic lethality associated with *Ubiad1* deficiency.

Although ubiquitination/ERAD-resistant HMGCR rescues embryonic lethality of *Ubiad1* deficiency (*Table 1*), *Ubiad1*$^{-/-}$: :*Hmgcr*$^{Ki/Ki}$ mice 8–12 weeks of age were 20–40% smaller than their *Ubiad1*$^{+/+}$: :*Hmgcr*$^{Ki/Ki}$ and *Ubiad1*$^{+/-}$: :*Hmgcr*$^{Ki/Ki}$ littermates (*Figure 3B and C*). It will be important in future studies to determine whether *Ubiad1*-deficient mice consume less food, have defects in nutrient absorption, or exhibit increased energy expenditure. UBIAD1 protein and its enzymatic product MK-4 were not detected in tissues of *Ubiad1*-deficient mice (*Figures 3B*, *4A and B*). However, it is important to note that despite the absence of MK-4, *Ubiad1*-deficient mice did not exhibit typical signs of vitamin K deficiency (*i.e.*, excessive hemorrhaging). The diet used in this study is supplemented with MD, which does not exhibit vitamin K activity (*Buitenhuis et al., 1990*) and is not converted to MK-4 in absence of UBIAD1. This indicates that the diet and/or gut microbiota provide sufficient amounts of vitamin K to support γ-glutamyl carboxylation of coagulation factors in *Ubiad1*$^{-/-}$: :*Hmgcr*$^{Ki/Ki}$ mice, suggesting their failure to thrive may result from disruption of carboxylation-independent activities of MK-4. *Figure 4B* shows that similar to previous results (*Harshman et al., 2016*; *Okano et al., 2008*), levels of MK-4 were highest in the pancreas and brain. Thus, failure of *Ubiad1*-deficient mice to thrive may result from reduced production of MK-4 in these or other tissues of the animals.

Recent studies show that inducible knockout of *Ubiad1* in adult mice caused death within 60 days (*Nakagawa et al., 2019*). The most striking abnormality of these mice was a remarkable reduction in pancreas size resulting from apoptotic disappearance of acinar cells. This observation prompted the authors to conclude that UBIAD1-mediated synthesis of MK-4 is essential for survival of pancreatic acinar cells. Our current studies reveal that the pancreas of *Ubiad1*$^{-/-}$: :*Hmgcr*$^{Ki/Ki}$ mice failed to produce MK-4 (*Figure 4B*); however, the organ was normal in size and exhibited no gross abnormalities (data not shown). These findings argue that ubiquitination/ERAD-resistant HMGCR allows for production of sterol and/or nonsterol isoprenoids distinct from MK-4 that are essential for subsistence of pancreatic acinar cells.

Blood chemistry analyses were conducted to determine whether *Ubiad1* deficiency results in damage of the liver and/or other organs. Compared to *Ubiad1*$^{+/+}$: :*Hmgcr*$^{Ki/Ki}$ littermates, *Ubiad1*$^{-/-}$: :*Hmgcr*$^{Ki/Ki}$ mice exhibited elevated levels of ALT (1.6-fold) and AST (>5 fold) in the serum (*Figure 4C and D*; *Figure 4—figure supplement 1*). Elevated levels of serum aminotransferases are routinely applied as biomarkers for hepatocyte injury. However, livers of *Ubiad1*-deficient mice did not feature gross abnormalities upon histological analysis (data not shown). Further examination revealed *Ubiad1* deficient mice exhibited a 2-fold increase in serum LDH and a 4–7.5-fold increase in CK that was accompanied by a 35% decrease in ALP (*Figure 4* and *Figure 4—figure supplement 1*). These observations are consistent with muscle injury and bone dysfunction in *Ubiad1*$^{-/-}$: :*Hmgcr*$^{Ki/Ki}$ mice. Indeed, degenerating skeletal muscle myofibers with macrophage infiltration and myofibers with centrally-localized nuclei as well as cell disorganization within the femoral growth plate were all observed by histological analysis of *Ubiad1*-deficient mice (*Figure 5*; *Figure 5—figure supplement 1* and **2**).

Statins, competitive inhibitors of HMGCR, are widely prescribed to lower plasma levels of cholesterol-rich low-density lipoprotein and reduce atherosclerotic cardiovascular disease. However, a significant fraction of patients undergoing statin therapy develop myopathy; a small portion of these patients progress to rhabdomyolysis (*Thompson et al., 2003*; *Ward et al., 2019*). Statin-induced myopathy has been attributed to depletion of mevalonate-derived metabolites resulting from inhibition of HMGCR. Skeletal muscle-specific knockout of HMGCR in mice causes severe myopathy that is rescued by mevalonate (*Osaki et al., 2015*). The observation that *Ubiad1*$^{-/-}$: :*Hmgcr*$^{Ki/Ki}$ mice exhibit signs of muscle injury suggests statin-induced myopathy may in part, result from MK-4 depletion. Support for this possibility requires MK-4 rescue experiments in *Ubiad1*-deficient mice and determination of whether *Hmgcr*$^{Ki/Ki}$ mice with skeletal muscle-specific knockout of *Ubiad1* develop myopathy.

Evidence indicates that vitamin K modulates bone homeostasis and metabolism through two mechanisms. One mechanism is mediated by osteocalcin and matrix Gla protein (*Fusaro et al.,*

*2017*), two VKDPs that play key roles in bone formation and mineralization. The second mechanism is mediated by the nuclear receptor known as steroid and xenobiotic receptor (SXR) in humans and pregnane X receptor (PXR) in mice. These promiscuous nuclear receptors are activated by a wide variety of xenobiotics and regulate genes involved in metabolism and clearance of the substances (*Kliewer, 2003*). *Pxr*-deficient mice present with osteopenia accompanied by reduced bone formation and increased bone resorption (*Azuma et al., 2010*). Considering that MK-4 has been reported to bind to and activate PXR (*Tabb et al., 2003*), it will be important to determine whether *Ubiad1*-deficiency phenocopies *Pxr*-deficiency with regard to bone homeostasis.

The characterization of genetically-manipulated mice underscores the physiological significance of UBIAD1 as an inhibitor of HMGCR ERAD. We recently generated mice (designated *Ubiad1^{Ki/Ki}*) harboring a knockin mutation that changes asparagine-100 (N100) to serine (N100S) (*Jo et al., 2019*). The N100S mutation in mouse UBIAD1 corresponds to SCD-associated N102S mutation in human UBIAD1 that abolishes sensing of membrane-embedded GGpp. UBIAD1 (N100S) was sequestered in ER membranes to inhibit ERAD of HMGCR, causing the protein's accumulation and overproduction of sterol and nonsterol isoprenoids in the liver and other tissues of *Ubiad1^{Ki/Ki}* mice. Significant corneal opacification was observed in *Ubiad1^{Ki/Ki}* mice greater than 50 weeks of age (*Hmgcr^{Ki/Ki}* mice used in the current study were not aged and thus, not examined for corneal opacification). Considered together with current studies, these findings unequivocally position UBIAD1 as a major regulator of HMGCR and mevalonate metabolism in vivo, provide new links between synthesis of sterols and MK-4, and establish *Ubiad1^{-/-}*: :*Hmgcr^{Ki/Ki}* mice as a model of MK-4 deficiency. Further analysis of these mice may reveal new physiological roles for MK-4 and additional pathways modulated by the vitamin K subtype.

# Materials and methods

## Key resources table

| Reagent type (species) or resource | Designation | Source or reference | Identifiers | Additional information |
|---|---|---|---|---|
| Genetic reagent (*M. musculus*) | Mouse/Wild Type:C57BL/6J | The Jackson Laboratory | Stock#000664 | |
| Genetic reagent (*M. musculus*) | Mouse/*Hmgcr^{Ki/Ki}* (HMGCR K89R/K248R):C57BL/6 | PMID: 27129778 | N/A | Knockin mice harboring mutations in the *Hmgcr* gene that prevent ubiquitination of HMGCR protein |
| Genetic reagent (*M. musculus*) | Mouse/*Ubiad1^{+/Δ172}*:C57BL/6J | This paper | N/A | Mice heterozygous for 172 bp deletion in exon 1 of the *Ubiad1* gene |
| Genetic reagent (*M. musculus*) | Mouse/Ubiad1^{Δ172/Δ172}: : *Hmgcr^{Ki/Ki}* (HMGCR K89R/K248R):C57BL/6J | This paper | N/A | *Hmgcr^{Ki/Ki}* mice homozygous for 172 bp deletion in exon 1 of the *Ubiad1* gene |
| Genetic reagent (*M. musculus*) | Mouse/*Ubiad1^{Δ29/Δ29}*: : *Hmgcr^{Ki/Ki}* (HMGCR K89R/K248R):C57BL/6J | This paper | N/A | *Hmgcr^{Ki/Ki}* mice homozygous for 29 bp deletion in exon 1 of the *Ubiad1* gene |
| Antibody | Rabbit monoclonal anti-SREBP-1 | PMID: 28244871 | IgG-20B12 | used at 1–5 µg/ml for immunoblots |
| Antibody | Rabbit monoclonal anti-SREBP-2 | PMID: 25896350 | IgG-22D5 | used at 1–5 µg/ml for immunoblots |
| Antibody | Rabbit polyclonal anti-UBIAD1 | PMID: 30785396 | IgG-205 | used at 1–5 µg/ml for immunoblots |
| Antibody | Rabbit polyclonal anti- HMGCR | PMID: 27129778 | IgG-839c | used at 1–5 µg/ml for immunoblots |
| Antibody | Rabbit polyclonal anti-Calnexin | Novus Biologicals | Cat#NB100-1965; RRID: AB_10002123 | used at 1–5 µg/ml for immunoblots |

*Continued on next page*

*Continued*

| Reagent type (species) or resource | Designation | Source or reference | Identifiers | Additional information |
|---|---|---|---|---|
| Antibody | Rabbit polyclonal anti-LSD-1 | Cell Signaling Technology | Cat#2139; RRID: AB_2070135 | used at 1–5 µg/ml for immunoblots |
| Sequence-based reagent | *Ubiad1* genotyping primers | Genotyping of mice is described in Materials and methods. | N/A | Forward: TCCCCTTGAGTGGCTCACTTTTA; Reverse: AAATCGAACAACATCCTGGGGCT |
| Sequence-based reagent | *Hmgcr*$^{Ki/Ki}$ genotyping primers | PMID: 27129778 | N/A | K89R Forward: GTCCATGAACATGTTCACCG; Reverse: CAGCACGTCCTATTGGCAGA K248R Forward: TCGGTGATGTTCCAGTCTTC; Reverse, GGTGGCAAACACCTTGTATC |
| Sequence-based reagent | Guide RNAs (gRNAs) used to target mouse Ubiad1 | Targeting of mouse *Ubiad1* gene is described in Materials and methods | N/A | gRNA-A: GGCTTCCCGAACGATCCTGG gRNA-B: CAAGTGCGCCTCCTACGTGT gRNA-C: TGTACACGGGGCCGGCAATT |
| Sequence-based reagent | qRT-PCR Primers for UBIAD1 | PMID: 30785396 | N/A | The sequence of these primers can be found in indicated reference |
| Sequence-based reagent | qRT-PCR Primers for SREBP1a | PMID: 30785396 | N/A | The sequence of these primers can be found in indicated reference |
| Sequence-based reagent | qRT-PCR Primers for SREBP-1c | PMID: 30785396 | N/A | The sequence of these primers can be found in indicated reference |
| Sequence-based reagent | qRT-PCR Primers for SREBP-2 | PMID: 30785396 | N/A | The sequence of these primers can be found in indicated reference |
| Sequence-based reagent | qRT-PCR Primers for HMGCR | PMID: 30785396 | N/A | The sequence of these primers can be found in indicated reference |
| Sequence-based reagent | qRT-PCR Primers for Insig-1 | PMID: 30785396 | N/A | The sequence of these primers can be found in indicated reference |
| Sequence-based reagent | qRT-PCR Primers for Insig-2a | PMID: 30785396 | N/A | The sequence of these primers can be found in indicated reference |
| Sequence-based reagent | qRT-PCR Primers for Insig-2b | PMID: 30785396 | N/A | The sequence of these primers can be found in indicated reference |
| Sequence-based reagent | qRT-PCR Primers for SCAP | PMID: 30785396 | N/A | The sequence of these primers can be found in indicated reference |
| Sequence-based reagent | qRT-PCR Primers for HMGCS | PMID: 30785396 | N/A | The sequence of these primers can be found in indicated reference |
| Sequence-based reagent | qRT-PCR Primers for FPPS | PMID: 30785396 | N/A | The sequence of these primers can be found in indicated reference |
| Sequence-based reagent | qRT-PCR Primers for LDLR | PMID: 30785396 | N/A | The sequence of these primers can be found in indicated reference |
| Sequence-based reagent | qRT-PCR Primers for PCSK9 | PMID: 30785396 | N/A | The sequence of these primers can be found in indicated reference |
| Sequence-based reagent | qRT-PCR Primers for ACS | PMID: 30785396 | N/A | The sequence of these primers can be found in indicated reference |
| Sequence-based reagent | qRT-PCR Primers for ACC1 | PMID: 30785396 | N/A | The sequence of these primers can be found in indicated reference |
| Sequence-based reagent | qRT-PCR Primers for FAS | PMID: 30785396 | N/A | The sequence of these primers can be found in indicated reference |
| Sequence-based reagent | qRT-PCR Primers for SCD1 | PMID: 30785396 | N/A | The sequence of these primers can be found in indicated reference |

*Continued on next page*

*Continued*

| Reagent type (species) or resource | Designation | Source or reference | Identifiers | Additional information |
|---|---|---|---|---|
| Sequence-based reagent | qRT-PCR Primers for GPAT | PMID: 30785396 | N/A | The sequence of these primers can be found in indicated reference |
| Sequence-based reagent | qRT-PCR Primers for LXRα | PMID: 30785396 | N/A | The sequence of these primers can be found in indicated reference |
| Sequence-based reagent | qRT-PCR Primers for ABCG5 | PMID: 30785396 | N/A | The sequence of these primers can be found in indicated reference |
| Sequence-based reagent | qRT-PCR Primers for ABCG8 | PMID: 30785396 | N/A | The sequence of these primers can be found in indicated reference |
| Sequence-based reagent | qRT-PCR Primers for GGPS | PMID: 30785396 | N/A | The sequence of these primers can be found in indicated reference |
| Sequence-based reagent | qRT-PCR Primers for Cyclophilin | PMID: 30785396 | N/A | The sequence of these primers can be found in indicated reference |
| Commercial assay or kit | TaqMan Reverse Transcription | Applied Biosystems | Cat#N8080234 | |
| Commercial assay or kit | Power SYBR Green PCR Master Mix | Applied Biosystems | Cat#4367659 | |
| Commercial assay or kit | DNeasy Blood and Tissue Kit | Qiagen | Cat#69506 | |
| Commercial assay or kit | MEGAshortscript Kit | Ambion | Cat#AM1354 | |
| Commercial assay or kit | Surveyor Mutation Detection Kit | Integrated DNA Technologies | Cat#706020 | |
| Commercial assay or kit | FuGENE6 Transfection Reagent | Promega | Cat#1815075 | |
| Chemical compound, drug | Menaquinone-4 | Sigma-Aldrich | Cat#809896 | |
| Chemical compound, drug | Phylloquinone (Vitamin K1) | Cerilliant | Cat#V-030 | |

## Mice

Previously described $Hmgcr^{Ki/Ki}$ mice harbor homozygous nucleotide mutations in the $Hmgcr$ gene that change lysine residues 89 and 248 to arginine (*Hwang et al., 2016*). These mutations prevent Insig-mediated ubiquitination and subsequent ERAD of HMGCR in the liver and other tissues of the knockin mice. $Ubiad1^{-/+}$ and $Ubiad1^{+/-}$: :$Hmgcr^{Ki/Ki}$ mice (C57BL/6N background) were generated using WT and $Hmgcr^{Ki/Ki}$ mice, respectively, using CRISPR/Cas9 technology in the Transgenic Core Facility at UT Southwestern Medical Center. The guide RNAs were designed to generate a deletion in exon 1 of the $Ubiad1$ gene, resulting in production of a truncated, nonfunctional protein (see *Figure 2*). $F_0$ founders were used to produce $F_1$ offspring that carried the $Ubiad1$-deficient allele through the germline. Pairs (male and female) of $Ubiad1^{+/-}$ and Ubiad1$^{+/-}$: :HmgcrKi$^{Ki/Ki}$ mice were intercrossed for production of homozygous $Ubiad1$-deficient mice in the WT or $Hmgcr^{Ki/Ki}$ background. To genotype $Ubiad1$-deficient animals, genomic DNA from tails was used for PCR with the primers indicated in the Key resources table against the mouse $Ubiad1$ sequence. The genotype of $Hmgcr^{Ki/Ki}$ mice was determined as described previously (*Hwang et al., 2016*). All mice were housed in colony cages with at 12 hr light/12 hr dark cycle and fed Envigo-Teklad Mouse/Rat Diet 2018 from Harlan Taklad (Madison, WI). All animal experiments were performed with the approval of the Institutional Animal Care and Use Committee at UT Southwestern Medical Center (APN - 2016–101636).

## Subcellular fractionation and immunoblot analysis

Approximately 80 mg of frozen tissue was homogenized in 500 µl buffer (10 mM HEPES-KOH, pH 7.6, 1.5 mM $MgCl_2$, 10 mM KCl, 5 mM EDTA, 5 mM EGTA, and 250 mM sucrose) supplemented with a protease inhibitor cocktail consisting of 0.1 mM leupeptin, 5 mM dithiothreitol, 1 mM PMSF, 0.5 mM Pefabloc, 5 µg/ml pepstatin A, 25 µg/ml N-acetyl-leu-leu-norleucinal, and 10 µg/ml aprotinin. The homogenates were then passed through a 22-gauge needle 10–15 times and subjected to

centrifugation at 1,000 X *g* for 5 min at 4°C. The 1,000 X *g* pellet was resuspended in 500 µl of buffer (20 mM HEPES-KOH, pH 7.6, 2.5% (v/v) glycerol, 0.42 M NaCl, 1.5 mM $MgCl_2$, 1 mM EDTA, 1 mM EGTA) supplemented with the protease inhibitor cocktail, rotated for 30 min at 4°C, and centrifuged at 100,000 X g for 30 min at 4°C. The supernatant from this spin was precipitated with 1.5 ml cold acetone at −20°C for at least 30 min; the precipitated material was collected by centrifugation, resuspended in SDS-lysis buffer (10 mM Tris-HCl, pH 6.8, 1% (w/v) SDS, 100 mM NaCl, 1 mM EDTA, and 1 mM EGTA), and designated the nuclear extract fraction. The post-nuclear supernatant from the original spin was used to prepare the membrane fraction by centrifugation at 100,000 X *g* for 30 min at 4°C. Each membrane fraction was resuspended in 100 µl SDS-lysis buffer. Protein concentration of nuclear extract and membrane fractions were measured using the BCA Kit (ThermoFisher Scientific). Prior to SDS-PAGE, aliquots of the nuclear extract fractions were mixed with 5X SDS-PAGE loading buffer to achieve a final concentration of 1X. Aliquots of the membrane fractions were mixed with an equal volume of buffer containing 62.5 mM Tris-HCl, pH 6.8, 15% (w/v) SDS, 8 M urea, 10% (v/v) glycerol, and 100 mM DTT, after which 5X SDS loading buffer was added to a final concentration of 1X. Nuclear extract fractions were boiled for 5 min, and membrane fractions were incubated for 20 min at 37°C prior to SDS-PAGE. After SDS-PAGE, proteins were transferred to Hybond C-Extra nitrocellulose filters (GE Healthcare, Piscataway, NJ). The filters were incubated with the antibodies described below and in the figure legends. Bound antibodies were visualized with peroxidase-conjugated, affinity-purified donkey anti-mouse or anti-rabbit IgG (Jackson ImmunoResearch Laboratories, Inc, West Grove, PA) using the SuperSignal CL-HRP substrate system (ThermoFisher Scientific) according to the manufacturer's instructions. Gels were calibrated with prestained molecular mass markers (Bio-Rad, Hercules, CA). Filters were exposed to film at room temperature. Antibodies used for immunoblotting to detect mouse SREBP-1 (rabbit monoclonal IgG-20B12), SREBP-2 (rabbit monoclonal IgG-22D5), HMGCR (IgG-839c), and UBIAD1 (rabbit polyclonal IgG-205) were previously described (*Engelking et al., 2005*; *Jo et al., 2011*; *McFarlane et al., 2014*; *Rong et al., 2017*). Rabbit polyclonal anti-calnexin IgG was purchased from Novus Biologicals (Centennial, CO). Rabbit polyclonal anti-LSD1 IgG was obtained from Cell Signaling (Danvers, MA). All antibodies were used at a final concentration of 1–5 µg/ml; the anti-calnexin antiserum was used at a dilution of 1:5000.

## Blood chemistry, MK-4 measurement, and histological analysis

Blood was drawn from the vena cava after mice were anesthetized in a bell jar atmosphere containing isoflurane. Serum was immediately separated and analyzed or stored at −80°C until use. Blood chemistries (cholesterol, triglycerides, AST, ALT, ALP, nonesterified fatty acids, etc.) were measured in the Metabolic Phenotyping Core Facility at UT Southwestern Medical Center.

MK-4 levels in mouse tissues was measured as follows. Approximately 100 mg of tissue from $Ubiad1^{+/+}$: :$Hmgcr^{Ki/Ki}$ and $Ubiad1^{-/-}$: :$Hmgcr^{Ki/Ki}$ mice was homogenized in phosphate-buffered saline (PBS) using a Powergen homogenizer (Fisher Scientific). The internal standard, vitamin $K_{1(25)}$, was added to homogenates generated from the kidney, pancreas, and spleen. The concentration of MK-4 in these homogenates was subsequently determined by reverse-phase HPLC as described previously (*Booth et al., 2008*) using a C30 column that allows improved resolution. The MK-4 content of the liver, brain, and adipose tissue was determined as described (*Fu et al., 2009*; *Harshman et al., 2016*) by LC-MS using deuterium-labeled vitamin $K_1$ as an internal standard.

The histological analysis of tissues from $Ubiad1^{+/+}$: :$Hmgcr^{Ki/Ki}$ and $Ubiad1^{-/-}$: :$Hmgcr^{Ki/Ki}$ mice was conducted by the Pathology Core at UT Southwestern Medical Center.

## Quantitative real-time PCR (qRT-PCR)

Total RNA was prepared from mouse tissues using the RNA STAT-60 kit (TEL-TEST 'B', Friendswood, TX). Equal amounts of RNA from individual mice were treated with DNase I (DNA-free, Ambion/Life Technologies, Grand Island, NY). First strand cDNA was synthesized from 10 µg of DNase I-treated total RNA with random hexamer primers using TaqMan Reverse Transcription Reagents (Applied Biosystems/Roche, Branchburg, NJ). Specific primers for each gene were designed using Primer Express software (Life Technologies) or Primer Bank of Harvard University. The real-time RT-PCR reaction was set up in a final volume of 20 µl containing 20 ng of reverse-transcribed total RNA, 167 nM of the forward and reverse primers, and 10 µl of 2X SYBR Green PCR Master Mix (Life

Technologies). PCR reactions were done in triplicate using ViiA7 Applied Biosystems. The relative amount of all mRNAs was calculated using the comparative threshold cycle ($C_T$) method. Mouse cyclophilin mRNA was used as the invariant control. Sequences for primers used for qRT-PCR are listed in the Key resources table.

## Quantification and statistical analysis

Graphs were generated, and statistical analysis was performed using Prism software (Graphpad). Quantitative data are presented as mean ± SEM. Statistical parameters (n, mean, SEM) can be found within the figure legends. The t-test was used to define differences between two datasets. The criterion for significance was set at $p < 0.05$. No statistical method was used to determine whether the data met assumptions of the statistical approach.

## Reproducibility of data

All results were confirmed in at least two independent experiments conducted on different days using different animals.

## Acknowledgements

We thank Dr. Guosheng Liang for helpful advice. This work was supported by National Institutes of Health grants HL-20948 (RD-B) and the USDA ARS Cooperative Agreement (58-1950-7-707) (SLB). Any opinions, findings, conclusions, or recommendations expressed in this publication are those of the authors and do not necessarily reflect the view of the USDA.

## Additional information

### Funding

| Funder | Grant reference number | Author |
| --- | --- | --- |
| National Institutes of Health | HL-20948 | Russell A DeBose-Boyd |
| U.S. Department of Agriculture | 58-1950-7-707 | Sarah L Booth |

The funders had no role in study design, data collection and interpretation, or the decision to submit the work for publication.

### Author contributions

Youngah Jo, Conceptualization, Data curation, Investigation, Visualization, Methodology, Writing - original draft, Writing - review and editing; Steven S Kim, Kristina Garland, Iris Fuentes, Lisa M DiCarlo, Investigation; Jessie L Ellis, Xueyan Fu, Resources, Investigation, Methodology; Sarah L Booth, Resources, Supervision, Funding acquisition, Investigation, Methodology, Project administration, Writing - review and editing; Bret M Evers, Resources, Supervision, Investigation, Methodology; Russell A DeBose-Boyd, Conceptualization, Supervision, Funding acquisition, Visualization, Methodology, Writing - original draft, Project administration, Writing - review and editing

### Author ORCIDs

Youngah Jo (iD) http://orcid.org/0000-0001-6779-3891
Bret M Evers (iD) http://orcid.org/0000-0001-5686-0315
Russell A DeBose-Boyd (iD) https://orcid.org/0000-0002-7452-5227

### Ethics

Animal experimentation: This study was performed in strict accordance with the recommendations in the Guide for the Care and Use of Laboratory Animals of the National Institutes of Health. All of the animals were handled according to approved institutional animal care and use committee (IACUC) protocols (#2016-101636) of the University of Texas Southwestern Medical Center.

Decision letter and Author response
Decision letter https://doi.org/10.7554/eLife.54841.sa1
Author response https://doi.org/10.7554/eLife.54841.sa2

## Additional files

### Supplementary files
• Transparent reporting form

### Data availability
All data generated or analysed during this study are included in the manuscript and supporting files.

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
