## [Decision Letter]

**Decision letter after peer review:**

Thank you for submitting your article "Enhanced ER-Associated Degradation of HMG CoA Reductase Causes Embryonic Lethality Associated with Ubiad1 Deficiency" for consideration by *eLife*. Your article has been reviewed by two peer reviewers, one of whom is a member of our Board of Reviewing Editors, and the evaluation has been overseen by a Reviewing Editor and Suzanne Pfeffer as the Senior Editor. The reviewers have opted to remain anonymous.

The reviewers have discussed the reviews with one another and the Reviewing Editor has drafted this decision to help you prepare a revised submission.

Summary:

This is a very focused study that aims to determine the reason behind embryonic lethality of germline deletion of Ubiad1 in mice. Earlier work from these authors had shown that Ubiad1, in addition to its role in producing MK-4, is an inhibitor of HMG-CoA-reductase (HMGCR) degradation under sterol replete conditions. The purpose of this inhibition is to spare a population of HMGCR for production of mevalonate, which is needed for the production of various downstream metabolites. Whether the lethality of Ubiad1 knockout is due to its role in MK-4 or its role in HMGCR protection was not known. This Research Advance addresses this question by knocking out Ubiad1 in mice that contain an HMGCR that is refractory to degradation. The key result is that these mice are viable, directly showing in a physiologic context that the role of Ubiad1 in protecting HMGCR is an important aspect of its function. A second important implication is that the mice now offer a system to more precisely investigate the consequences of failed production of MK-4. This study directly follows from the authors' earlier line of investigation in two previous *eLife* papers and represents an important advance. The referees raised points that should be addressable with clarifications, changes to the text/figures, and possibly straightforward experiments.

Essential revisions:

1) The authors state in the Results section that hepatic cholesterol and triglycerides were not significantly changed. However, in Figure 3—figure supplement 1, triglycerides (and possibly also cholesterol) seems to be different in females. Similarly, Insig2a and -2b also seems to be different. Perhaps a more nuanced interpretation of these results is warranted. The authors should comment on these changes (unless they are really not significant despite the small error bars).

2) The authors explain in the Discussion section that AST and ALT are classic markers of liver injury. This should be explained when these enzymes are first mentioned so a reader without this knowledge is not confused.

3) Missing word: "The second mechanism is mediated the nuclear receptor…" Discussion section.

4) The authors previously showed that the SCD mutant Ubiad1 leads to cholesterol overload and corneal opacifications. One might expect that the ERAD-resistant HMGCR would show a similar phenotype. Is this observed (or has it been seen in earlier analyses)? Can the authors comment on this near the end of their Discussion? Perhaps the mice were not aged enough?

5) In Figure 3—figure supplement 1, the bar graphs in panels B and C would benefit by an indicator of statistical significance where applicable. Some of the differences seem significant by eye, but a statistical test is not indicated.

6) In Figure 3 it is unclear why HMGRwt/wt and ki/ki animals are shown separately (in panels A and B). Direct comparison between these animals would be very valuable (for example to assess the effect of the ki allele on HMGR levels and SREBP processing.

7) In Figure 3B, the HMGR levels in the various UBIAD1 genotypes appear to change (and also SREPB1). Even more strangely, they follow opposite trends in male and female mice. The data also contrasts with the one for Allele B (shown in Figure 3—figure supplement 1A). It would be good to clarify the inconsistencies.

8) Ubiad1-/- HmgcrKi/Ki mice are born at Mendelian ratios but display a significant growth phenotype however, this is only characterized very superficially. Simple histological analysis showed abnormalities in muscles and bone but how these correlate with the growth defect is unclear. Do these mice eat less or absorb less nutrients? Do they expend more energy to the point of inducing muscle lesions? I understand that some of these issues fall beyond the scope of this manuscript but they should at least be discussed.

9) In the Results section, the data on the growth defect of Ubiad1-/- HmgcrKi/Ki is presented twice (at 7.7 and 8 weeks) with slightly different numbers. Similarly, it is unclear why some of the blood analysis (for example AST and ALT) are shown in duplicate in Figure 4C and D. These redundancies are unnecessary.

---

## [Author Response]

Summary:This is a very focused study that aims to determine the reason behind embryonic lethality of germline deletion of Ubiad1 in mice. Earlier work from these authors had shown that Ubiad1, in addition to its role in producing MK-4, is an inhibitor of HMG-CoA-reductase (HMGCR) degradation under sterol replete conditions. The purpose of this inhibition is to spare a population of HMGCR for production of mevalonate, which is needed for the production of various downstream metabolites. Whether the lethality of Ubiad1 knockout is due to its role in MK-4 or its role in HMGCR protection was not known. This Research Advance addresses this question by knocking out Ubiad1 in mice that contain an HMGCR that is refractory to degradation. The key result is that these mice are viable, directly showing in a physiologic context that the role of Ubiad1 in protecting HMGCR is an important aspect of its function. A second important implication is that the mice now offer a system to more precisely investigate the consequences of failed production of MK-4. This study directly follows from the authors' earlier line of investigation in two previous eLife papers and represents an important advance. The referees raised points that should be addressable with clarifications, changes to the text/figures, and possibly straightforward experiments.Essential revisions:1) The authors state in the Results section that hepatic cholesterol and triglycerides were not significantly changed. However, in Figure 3—figure supplement 1, triglycerides (and possibly also cholesterol) seems to be different in females. Similarly, Insig2a and -2b also seems to be different. Perhaps a more nuanced interpretation of these results is warranted. The authors should comment on these changes (unless they are really not significant despite the small error bars).

In the revised manuscript, we note that Insig-2a mRNA and levels of HMGCR and SREBPs vary in the *Ubiad1*-deficient mice. The nature of these variations is not clear, but may be related to the failure of the animals to thrive. This is now discussed in the text.

2) The authors explain in the Discussion section that AST and ALT are classic markers of liver injury. This should be explained when these enzymes are first mentioned so a reader without this knowledge is not confused.

We have now indicated in the Discussion section that AST and ALT are classic markers of liver injury.

3) Missing word: "The second mechanism is mediated the nuclear receptor…" Discussion section.

This error has now been corrected. We thank the reviewer for pointing this out.

4) The authors previously showed that the SCD mutant Ubiad1 leads to cholesterol overload and corneal opacifications. One might expect that the ERAD-resistant HMGCR would show a similar phenotype. Is this observed (or has it been seen in earlier analyses)? Can the authors comment on this near the end of their Discussion? Perhaps the mice were not aged enough?

The reviewer is correct, corneal opacification would only be expected to occur in aged *Hmgcr^Ki/Ki^*mice. In these studies, the mice were not aged and thus, not examined for corneal opacification. This is now indicated in the Discussion section.

5) In Figure 3—figure supplement 1, the bar graphs in panels B and C would benefit by an indicator of statistical significance where applicable. Some of the differences seem significant by eye, but a statistical test is not indicated.

We included in the original manuscript a graph for hepatic triglycerides that included incorrect error bars. In the corrected graph, the differences are not statistically different. The only gene whose expression is substantially different in the Ubiad1 deficient mice is Insig-2a. We have now commented on this in the revised manuscript.

6) In Figure 3 it is unclear why HMGRwt/wt and ki/ki animals are shown separately (in panels A and B). Direct comparison between these animals would be very valuable (for example to assess the effect of the ki allele on HMGR levels and SREBP processing.

In previous studies (Hwang et al., 2016 and Hwang et al., 2017), we established that the knockin allele causes marked accumulation of HMGCR protein levels and reduces processing of SREBPs. In all of our studies, comparisons are made between litter mates. Here, the studies were conducted independently, and importantly, animals harboring wild type and the knockin allele are not littermates.

7) In Figure 3B, the HMGR levels in the various UBIAD1 genotypes appear to change (and also SREPB1). Even more strangely, they follow opposite trends in male and female mice. The data also contrasts with the one for Allele B (shown in Figure 3—figure supplement 1A). It would be good to clarify the inconsistencies.

Please see response to point 1.

8) Ubiad1-/- HmgcrKi/Ki mice are born at Mendelian ratios but display a significant growth phenotype however, this is only characterized very superficially. Simple histological analysis showed abnormalities in muscles and bone but how these correlate with the growth defect is unclear. Do these mice eat less or absorb less nutrients? Do they expend more energy to the point of inducing muscle lesions? I understand that some of these issues fall beyond the scope of this manuscript but they should at least be discussed.

We have now discussed this issue in the revised manuscript.

9) In the Results section, the data on the growth defect of Ubiad1-/- HmgcrKi/Ki is presented twice (at 7.7 and 8 weeks) with slightly different numbers. Similarly, it is unclear why some of the blood analysis (for example AST and ALT) are shown in duplicate in Figure 4C and D. These redundancies are unnecessary.

The experiment shown in Figure 3B shows weight of the Ubiad1-deficient animals after 8 weeks. In Figure 3C, we wanted to compare weight gain of the animals post-weaning to demonstrate the Ubiad1-deficient animals exhibited growth defects. In Figure 4C and 4D, different sets of liver enzymes were measured in the experiments.